

# Assessment of Social Vulnerability to Natural Hazards in Nepal

Dipendra Gautam[1]

[1]Structural and Earthquake Engineering Research Institute, Kathmandu, Nepal

*Correspondence to*: Dipendra Gautam (dipendra.gautam.seri@gmail.com)

**Abstract.** This paper disseminates district-wise social vulnerability to natural hazards in Nepal. Disasters like earthquake, flood, landslide, epidemic and drought are common in Nepal. Every year thousands of people are killed along with huge economic and environmental losses are reported in Nepal due to various natural disasters. Although natural hazards are well recognized in Nepal, quantitative as well as qualitative social vulnerability mapping does not exist until now. This study aims to fulfil the hiatus of such quantification considering district level social vulnerability to natural hazards using available census.
To perform district level vulnerability mapping, 13 variables were selected and aggregated indexes were plotted in Arc GIS environment to depict the level of social vulnerability for all 75 districts in Nepal. Only 4 districts were found to be under very low social vulnerability whereas 46 districts (61%) were found to be under moderate to high social vulnerability status. Vulnerability mapping highlights the immediate need for decentralized frameworks to tackle natural hazards in district level and the results of this study can contribute to preparedness, planning and resource management, inter-district coordination,
contingency planning and public awareness efforts.

## 1 Introduction

Nepal is characterized by frequent occurrence of natural disasters. Geo-Seismotectonic setting, annual torrential precipitation, climate change impacts, among others are the leading causes of natural disasters in Nepal. Notably, the first decade of 21st century was contemptible for Nepal due to loss of above 15000 people, thousands were injured and property losses were
enormous due to anthropogenic disasters in terms of armed conflict. Apart from this, multi-faceted disasters occur almost every year leading to enormous losses in terms of socio-economic and environmental sectors. Nepal is ranked 20th for multi-hazard proneness; 4th in case of climate change; 11th in case of earthquake hazard and 30th in terms of flood hazards (UNDP/BCRP 2004). Recent events like the 2009 flood in eastern Nepal; 2011 earthquake in eastern Nepal and Gorkha earthquake (2015) justify the occurrence of frequent and devastating mega-disasters in Nepal. Although it is well known in Nepal that country is
under severe exposure level of multi-hazards, multi-hazard risk assessment is not performed yet thus proper designation in terms of hazard level to various parts of the country is not available.



The overall risk due to natural hazards depends on hazard (H), vulnerability (V), resources (Re), counter interventions for identified hazards (Ci), exposure to hazard (E) and perception by the stakeholders (P) can be depicted as follows:

$$R = H \times (V - Re - Ci) \times E \times P \qquad (1)$$

Endorsement of hazards and associated studies were incepted in Nepal since 1982 when the Natural Calamity Relief Act (1982) was formulated for the first time in South Asia and before majority of the countries in the world developed their disaster risk planning. Even though basic endorsement started in early 1980s, dynamics in later dates was not as per expected level. Thus, losses were continued and no subsequent amendments can be seen to the act till date. In policy level, endorsement of building code since 2003 became the first major intervention to counteract the earthquake hazard however implementation of building code remains confined to some urban centers of Nepal and large part of country remains under severe threat of devastation in case of strong earthquake. This is consistent with the damage during 2015 Gorkha earthquake wherein virtually all collapse or damage was attributed to rural neighborhoods of central and eastern Nepal with poor construction practices (Gautam and Chaulagain 2016; Gautam et al. 2016). Limited works related to earthquake and landslide hazard mapping are done in local to regional scales in Nepal. Chaulagain et al. (2015) assessed seismic risk and mapped the seismic hazard across Nepal. Similarly, Paudyal et al. (2012), Gautam and Chamlagain (2016) and Gautam et al. (2017) performed local scale hazard analyses and developed microzonation maps. In addition to this, Chaulagain et al. (2016) performed loss estimation assessment in case of earthquakes for Kathmandu valley. In terms of landslides, Devkota et al. (2013) developed landslide susceptibility maps in regional scale whereas studies related to other catchments and regions do not exist in contemporary literatures. After 2000, earthquake is mostly discussed topic in Nepal in policy to local level. Meanwhile landslides, floods and other hazards are not given equal importance in policy level and academic researches. Existing literatures and works have not covered social vulnerability to natural hazards even though risk perception has reached up to public and awareness is exponentially increased in almost all settlements of Nepal. It is worthy to note here, the awareness noted above is limited to earthquakes only and other hazards are not perceived as devastating as earthquake in public level. Centralized and urban-concentrated resource allocation practice is still becoming perilous to the public of remote locations of Nepal as reinforced by the evidences after the Gorkha earthquake; people in the remote locations were not reached for several weeks after the main shock and whereabouts of thousands of people was under dilemma. Most of the urban as well as rural settlements are exposed to multi-hazards, in this context, social vulnerability analysis and mapping is a dire need for Nepal. Such mapping can have direct influence in policy making to preparedness activities. Apart from this, even ordinary people could perceive the level of vulnerability in map. Social vulnerability index has gained momentum worldwide since its inception and successful implementation in different locations worldwide. For example, Blaikie and Brookfield (1987), Chambers (1989), Dahl (1991), Cutter et al. (1997), Balaikie et al. (1994), Mileti (1999), Morrow (1999), King and MacGregor (2000) and Cutter et al. (2003) among others provided strong background and motivation for development and implication of social vulnerability index. After 2005, intensive focus has been provided in construction and mapping of social vulnerability index (e.g. de Oliviera Mendes (2009), Wood et al. (2010), Bjarnadottir et al. (2011), Holand et al. (2011), Yoon (2012), Armas and Gavris (2013), Lixin et al. (2013), Guillard-Gonçalves et al. (2013), Siagian et al. (2014), Garbutt et al. (2015), Hou et al. (2015), de Loyola Hummell et al. (2016), Frigerio





and de Amicis (2016), Roncancio and Nardocci (2016)). On the contrary, no any vulnerability assessments exist in literature even though natural hazards are frequent due to tectonic setting, annual torrential precipitation, steep topography, climate change and unsustainable and haphazard construction practices and lack of basic health facilities. In addition, Nepal's preparedness and policy interventions are way behind when compared to the existing hazard, exposure and perception level.

To fulfill gap between exposure and preparedness, this study depicts district level social vulnerability mapping based on vulnerability scores calculated from selected variables.  After all, some suggestions are made for policy, preparedness and future way forwards.

## 2 Materials and Methods

Nepal does not update the database for population, households, infrastructures, facilities and others regularly. Moreover, digital

database is limited thus only the censuses are the reliable data sources. Even in case of census, the coverage in terms of variables is largely constrained to population categories and more specific data like single year population, per capita income in local level, village level census is still lacking until 2011 census however although 2011 census progressed appreciably when compared to 2001 census. Present study is based on 2011 census as reported by the Central Bureau of Statistics (CBS) under National Planning Commission (NPC) of Nepal (CBS 2012). Both 2011 and 2001 census were used for 13 variables

used in this study. Only 13 variables were used in this study adapting reliable and available ones as most of the information were not strictly associated with social vulnerability to natural hazards. Table 2 depicts the description of variables used in this study along with cardinality of each variable. Broadly, social vulnerability assessment can be categorized under two approaches as: a) deductive and b) inductive. Deductive approach is based on selection of limited variables as done by Cutter et al. (2000), Wu et al. (2002), Zahran et al. (2008) and others. Meanwhile inductive approach uses more organized and exhaustive social

vulnerability assessment framework with all possible variants considered at a time. Recent advances in social vulnerability assessment is more focused towards inductive approach due to availability of database (e.g. Cutter et al. 2003; de Loyola Hummel et al. 2016). Detailed comparison between deductive and inductive approaches is reported by Yoon (2012). As of recent trend, social vulnerability index (SoVI) mapping is undoubtedly superior to generalized score based vulnerability mapping but such mapping needs many variables and that is not feasible for Nepal. Thus, a generalized deductive approach

with standardized individual vulnerability score was calculated for limited variables and then integrated to depict the social vulnerability level. Under this framework, each variable was converted into a common scale using maximum value transformation approach as used by Cutter et al. (2000). In this approach a ratio between the value of a variable to the maximum value of the same is calculated as:

$$Score\ (N_i) = \frac{value\ of\ variable\ i}{maximum\ value}$$





As noted by Cutter et al. (2000), higher value of score signifies higher vulnerability. After normalization of all variables in between 0 and 1, the social vulnerability index was calculated for each district by integrating the scores of each variable per cardinality as:

$$Total\ Vulnerability\ Score\ (TVS) = \sum_{i=1}^{13} N_i$$

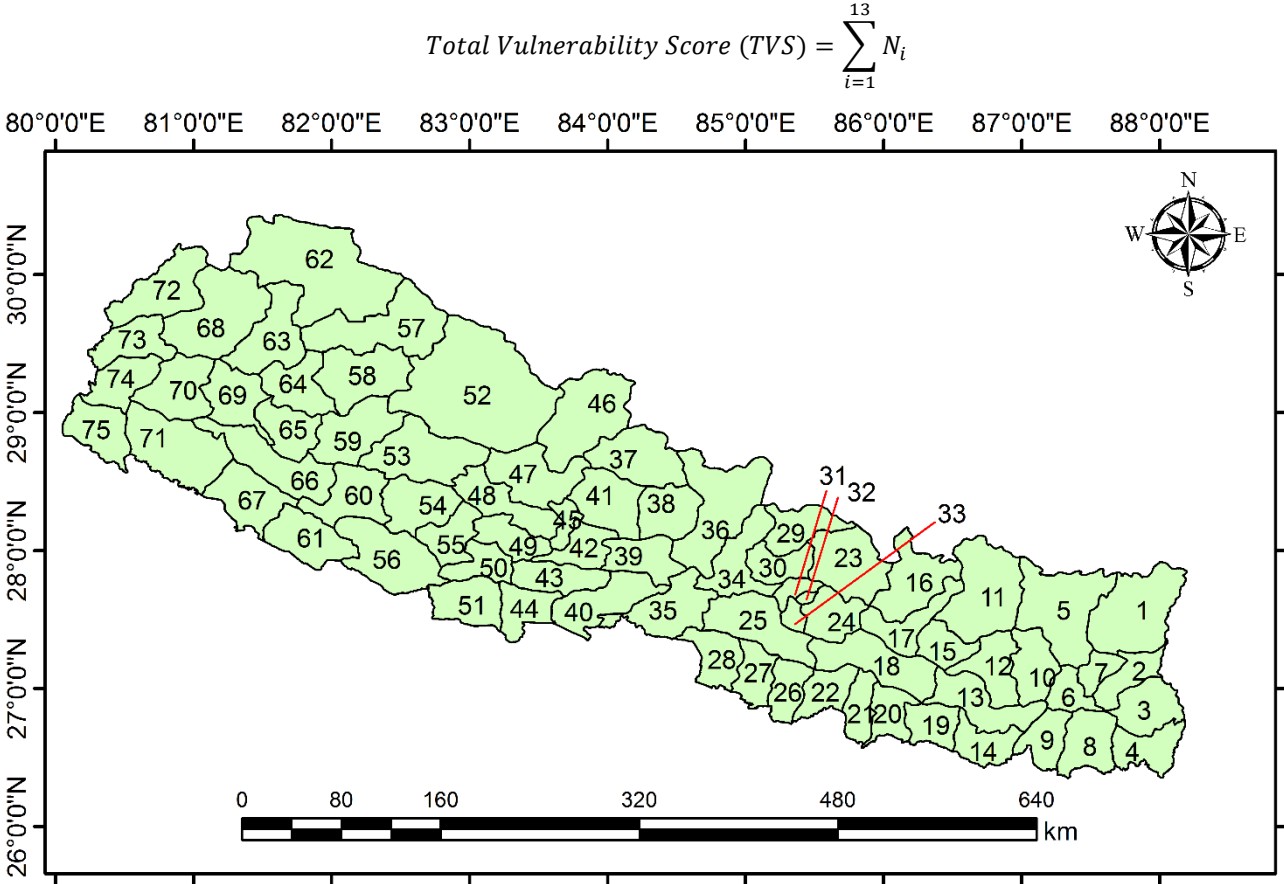

**Fig. 1** Districts in Nepal, map serials indicate: 1. Taplejung 2. Panchthar 3. Ilam 4. Jhapa 5. Sankhuwasabha 6. Dhankuta 7. Tehrathum 8. Morang 9. Sunsari 10. Bhojpur 11. Solukhumbu 12. Khotang 13. Udaypur 14. Saptari 15. Okhaldhunga 16. Dolakha 17. Ramechhap 18. Sindhuli 19. Siraha 20. Dhanusha 21. Mahottari 22. Sarlahi 23. Sindhupalchowk 24. Kavre 25. Makwanpur 26. Bara 27. Rautahat 28. Parsa 29. Rasuwa 30. Nuwakot 31. Kathmandu 32. Bhaktapur 33. Lalitpur 34. Dhading 35. Chitwan 36. Gorkha 37. Manang 38. Lamjung 39. Tanahun 40. Nawalparasi 41. Kaski 42. Syangja 43. Palpa 44. Rupandehy 45. Parbat 46. Mustang 47. Myagdi 48. Baglung 49. Gulmi 50. Arghakhachi 51. Kapilvastu 52. Dolpa 53. Rukum 54. Rolpa 55. Pyuthan 56. Dang 57. Mugu 58. Jumla 59. Jajarkot 60. Salyan 61. Banke 62. Humla 63. Bajura 64. Kalikot 65. Dailekh 66. Surkhet 67. Bardiya 68. Bajhang 69. Achham 70. Doti 71. Kailali 72. Darchula 73. Baitadi 74. Dadeldhura 75. Kanchanpur



**Table. 1** Variables used in SoVI analysis

| Variable name | Cardinality | Description |
|---|---|---|
| N1 | + | % of households without telephone service |
| N2 | - | % of population with cellular phone service |
| N3 | + | % of households without at least one means of information services (TV, internet, radio) |
| N4 | + | % of females |
| N5 | + | Population density |
| N6 | + | % of female-headed households with no shared responsibility |
| N7 | + | Average no. of people per household |
| N8 | + | Average no. of people illiterate aged 5 and above |
| N9 | + | Population change (2000-2010) |
| N10 | + | % of people with at least one disability |
| N11 | + | % of population under age 14 and over 60 |
| N12 | + | % of households with no toilet |
| N13 | + | % of house with no electricity services |

Finally, the social vulnerability indexes were classified into five different classes based on standard deviation as shown in Table 2. As per the convention depicted in Table 2, Arc GIS mapping was done for each district in terms of vulnerability level to generate a thematic map that highlights the distribution of social vulnerability to natural hazards in Nepal.

5   **Table 2**. Vulnerability level classification based on standard deviation

| Standard deviation ($\sigma$) | Level of vulnerability |
|---|---|
| >1.5$\sigma$ | Very high |
| 0.5 – 1.5$\sigma$ | High |
| -0.5 – 0.5$\sigma$ | Moderate |
| -1.5 - -0.5$\sigma$ | Low |
| < -1.5$\sigma$ | Very low |

## 3 Results and Discussion

SoVI scores were calculated for all 75 districts by integration of individual variable scores. Table 3 presents the descriptive statistics of each of the variable used for social vulnerability analysis. As shown in Table 3 the variance in most of the variables
10  is generally high. This is due to widespread discrepancies between the districts in terms of social structure, economic development, infrastructural development, basic life services and access. Districts in central and eastern regions of Nepal are



more developed than the districts in western regions of Nepal. In addition to this, the facilities are concentrated in urban centers of Kathmandu valley and southern Indo-Gangetic plains. For instance, the telephone access is concentrated to only urban centers thus mountainous districts are not well reached with this service. Apart from this, the armed conflict between 1996 and 2006 led isolation of most of the mountainous districts specifically in the western mountainous areas. Similarly, the cellular

phone service was opened to public only after 2006 and this facility was concentrated to major urban centers and southern Indo-Gangetic plains until 2010. However, in later dates the reach has become far better as highlighted by the variable (N2). Information and communication is very important aspect for rapid response and life safety. For example, the 2012 flood of Seti river in western Nepal was instantly broadcasted to the downstream people thus the losses were far less than expected. Communication systems in Nepal are also concentrated to urban areas, Indo-Gangetic plains and up to middle mountains

leaving behind the high mountains and western mountains far behind thus variance is observed to be very high. Almost all districts in Nepal have higher population of females than males. It is associated with the social norms in terms of importance of son to continue the future generation. This is also partly backed by reach of education in eastern and central mountains and people started using family planning tools and started to give birth to fewer child than the western mountain peoples. This concept has been eradicated in major urban centers however southern plains and mountains have not progressed much hence

female population is very high in such areas. The sparse distribution of population in mountains is surprisingly low due to migration towards the areas with better facilities. In case of eastern and central mountains the population change between 2000 to 2010 is negative leading to negative population growth rate. The armed conflict was in its peak during 2000 to 2006 thus people migrated to urban areas where security was assured. Due to tough social provisions imposed by the rebels, the western mountain districts did not follow the same trend as that of eastern mountains thus western mountains maintained positive

population growth. Kathmandu district (serial 31 in Fig. 1) alone has population density of 4416 people per square kilometers. On the contrary, Manang district (serial 37 in Fig. 1) has population density of 3 persons per square kilometers. Percentage of female headed population in Nepal started increasing after 2000 due to change in the social norms that were associated with male's supremacy. Government of Nepal offered discounts on land title transfer tax if the title is changed to females thus the trend is ever increasing in recent dates. Natural disasters have historically suffered worst to the females in Nepal. For example,

Gorkha earthquake of 2015 April 25 killed more females than males. This is because females in Nepal are mostly confined to household activities and remain inside their homes during the events. Similar observations were made during the floods in the southern plains at various times.

**Table 3.** Descriptive statistics of variables considered for social vulnerability assessment

| Variables | Standard deviation | Mean | Max. | Min. |
|-----------|--------------------|------|------|------|
| N1 | 5.45 | 95.18 | 99.39 | 69.62 |
| N2 | 4.24 | 12.17 | 22.66 | 3.93 |
| N3 | 10.96 | 18.53 | 45.79 | 1.66 |



| N4  | 2.34   | 51.98 | 56.81 | 44    |
|-----|--------|-------|-------|-------|
| N5  | 585.69 | 312   | 4416  | 3     |
| N6  | 4.28   | 8.67  | 17.7  | 1.20  |
| N7  | 0.62   | 4.90  | 6.44  | 3.92  |
| N8  | 8.96   | 32.14 | 54.35 | 12.14 |
| N9  | 14.80  | 9.72  | 61.23 | -31.8 |
| N10 | 0.90   | 2.43  | 5.39  | 0.9   |
| N11 | 4.69   | 44.48 | 52.79 | 29.8  |
| N12 | 20.84  | 39.77 | 79.26 | 0.85  |
| N13 | 25.73  | 42.98 | 95.98 | 1.88  |

Average number of people per household varies Nepal due to geographical locations and cultural groups. In remote locations, the child birth rate is usually high thus average number of people per household is high. However, in case of urban neighborhoods multiple families share a single building (either joint family or rented family). The census lacks specific

information regarding the rented families thus it was not possible to classify and define a separate variable for this aspect. Nepal has progressed considerably in education sector specially after 2000. Students from marginalized communities, ethnic minorities, certain geographical locations are given stipends and reservations for basic and higher education. Although the educational status is not comparable to developed states till now. People who could read and write in Nepali language are defined as literate in Nepal. As per the 2011 census, literacy status in some of the southern plain districts and western mountain

districts was very low. This is due to social problems like the value of education in community level, early marriage and high dependence in subsistence farming. Population with at least one deficiency varies between 0.9 to 5.39% in Nepal. Nepal is striving for basic health facilities until now. Majority of the health facilities, preventive measures and children vaccinations were started after the restoration of democracy in 1990 thus still considerable fraction of population has at least one deficiency. Till now Nepal eradicated Polio and Malaria and progressed in controlling other diseases too. The variation of economically

unproductive population (population below 15 years and above 60 years) ranges from 29.8 to 52.79% in Nepal. This denotes dependent population is very high and impact of disaster is particularly intense in such groups. Sanitation is still big issue for Nepalese people. Percentage of households without toilets are still up to 79.26%. In addition to this, clean drinking water is not assured to each household in Nepal. The various sources of water supply like tap water, springs and others are not independently verified in terms of water quality. Apart from this, thousands of people suffered from water-borne diseases are

reported during every spring and monsoon in Nepal. The 2011 census consists various water supply resources for households in Nepal however due to water quality issue this variable was not considered for assessment of social vulnerability index.

The social vulnerability indexes for each district per generalized conventions based on standard deviation are depicted in Fig. 2. As shown in Fig. 2, social vulnerability to natural hazards is higher in western mountains than the eastern and central regions



of Nepal. Similarly, only four districts are under very low vulnerability level in Nepal. This scenario depicts higher vulnerability to natural hazards nationwide. Western Nepal is long identified as potential hotspot of future mega-earthquake, famine and epidemics thus instant interventions are required to tackle the very high to high vulnerability status of districts in this region.

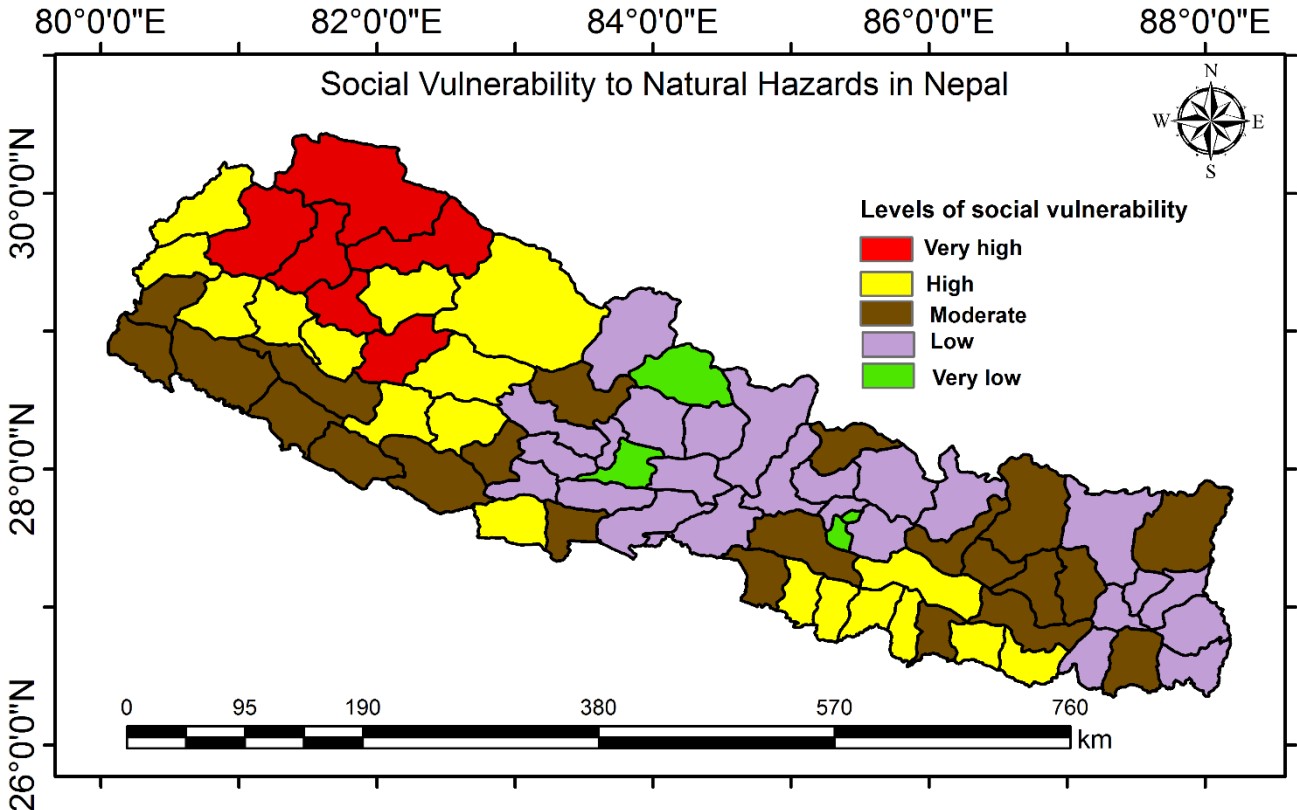

**Fig. 2** District-wise social vulnerability to natural hazards in Nepal

Frequency of districts in terms of social vulnerability level is outlined in Table 4. As shown in Table 4, 6 (8%) out of 75 districts are under very high social vulnerability level, 18 (24%) districts are under high vulnerability level, 22 (29.33%) districts are under moderate social vulnerability, 25(33.33%) districts are under low vulnerability level and 4 (5.34%) districts are under very low vulnerability level in Nepal. Results show that 32% of districts are under high vulnerability level thus District Natural Disaster Relief Committee (DDNRC) are needed to be strengthened with adequate resources in these districts more than others. As observed during the Gorkha earthquake of 2015, Sindhupalchowk landslide of 2014 and Koshi flood of 2008; every relief, response and recovery effort was governed by the Central Natural Disaster Relief Committee (CNDRC) that led to delayed response and sometimes hindered by the weather extremities too. In addition to this, it was observed that Local Natural Disaster Relief Committee (LNDRC) were completely defunct during these events thus preparedness in terms of uplifting local committees is direly needed in Nepal. Even after the federal states become functional; districts will not be



changed thus contingency planning to sustainable natural disaster preparedness initiatives are urgent specially for western mountains. Resource allocations, training first responders, district level planning and overall budget allocation can have benefit of the mapping done in this study.

**Table 4.** Frequency of districts in terms of social vulnerability level

| Level of social vulnerability | Number of districts | Districts |
|---|---|---|
| Very high | 6 | Jajarkot, Kalikot, Mugu, Humla, Bajura, Bajhang |
| High | 18 | Siraha, Saptari, Mahottari, Sarlahi, Sindhuli, Rautahat, Bara, Kapilvastu, Rolpa, Rukum, Salyan, Dailekh, Dolpa, Jumla, Achham, Doti, Baitadi, Darchula |
| Moderate | 22 | Taplejung, Morang, Bhojpur, Solukhumbu, Okhaldhunga, Khotang, Udaypur, Dhanusha, Ramechhap, Rasuwa, Makwanpur, Parsa, Myagdi, Rupandehy. Pyuthan, Dang, Banke, Bardiya, Surkhet, Kailali, Kanchanpur, Dadeldhura |
| Low | 25 | Panchthar, Ilam, Jhapa, Sunsari, Dhankuta, Tehrathum, Sankhuwasabha, Dolakha, Sindhupalchowk, Kavre, Kathmandu, Nuwakot, Dhading, Chitwan, Gorkha, Lamjung, Tanahun, Kaski, Mustang, Parbat, Baglung, Gulmi, Palpa, Nawalparasi, Arghakhachi |
| Very low | 4 | Lalitpur, Bhaktapur, Manang, Syangja |

Being a multi-hazard prone country, multi-hazard mapping is urgently needed so that social vulnerability mapping could be integrated with the multi-hazard maps to depict highly precise thematic maps. Nepal lacks highly classified researches regarding hazard mapping; it is due to lack of national interest and academic focus. It is obvious that if national priority is considered in specific hazard to multi-hazard mapping across the country responding natural disasters would be much easier

10 in terms of policies to in-situ interventions. In addition to this, data management, digitization and coverage of more variables during census will increase the quality of social vulnerability indexes thus future censuses should consider for more variables. Finally, local constituencies below the district level are being formulated as the primary units of Federal Democratic Republic of Nepal thus local constituency level social vulnerability mapping will be more effective than district level mapping if database can be organized immediately after formulation of such constituencies.

15 **4 Conclusion**

This study is the first attempt to understand the district level vulnerability in Nepal. The social vulnerability score is calculated and mapped for all 75 districts of Nepal. Being a natural disaster prone country, Nepal needs to develop effective mitigation,



prevention and contingency plans for all potential natural hazards so this study could be fundamental for the policy makers and stakeholders to initiate interventions in district level. Results have confirmed that western mountain districts are under very high to high social vulnerability status whereas eastern and central regions depicted low to moderate social vulnerability to natural disasters in general.  Losses due to natural hazards in western mountain districts would be very high in case of major

natural hazards thus immediate actions are needed. Previous natural disasters have reflected a poor coordination, delayed response and marginal preparedness scenario from the central level. Thus, decentralization in terms of preparedness, response and recovery is necessary for Nepal because of the district wise variation of social vulnerability to natural hazards.

In social vulnerability assessment, data constraint plays important role and thus affects the results. Adaptation of more variables is important to assure precision and proper representation of social vulnerability. In addition to this, spatial variation within a

district has also remarkable influence as vulnerability mapping considers uniform variation of values of variables across the districts that is not strictly per the ground conditions. To overcome this, local level social vulnerability mapping should be considered in future. Apart from this, exhaustive and more reliable social vulnerability index (SoVI) mapping based on principal component analysis is needed for Nepal and this approach should be used in future works.

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
