# Peer review of "Assessment of Social Vulnerability to Natural Hazards in Nepal"

_Natural Hazards and Earth System Sciences, 2017_

## Referee Comment (RC1) · Anonymous Referee #1 · 15 May 2017

This is a good study - it has the basis for becoming a good article as overall it is well written - although language needs to be thoroughly revised.

However there are a few key points to consider which is why I am recommending a major revision: 1. I am concerned that your variables do not consider post EQ situation as well as climate change variables which are really creating higher vulnerability for several districts (esp Mustang and Manang which have surprising low vulnerability considering their current situation) and that are not reflected in your results... I encourage you to take another look at your data and see how to incorporate the post EQ situation as well as multi risk... 2. If you are not able to revise your data analysis to incorporate the above mentioned parameters, you really need to develop a much more critical synthesis in your conclusions with a critique of SOVI which does not go far enough in incorporating multi risk nor the post EQ situation.... 3. Your

vulnerability formula is not reflected in the results and conclusions - i.e. where do you reflect on perception and resources? Why did you select this formula- where does it come from ? 4. It is not fair to assert that there are no social vulnerability studies from Nepal - your study would be made more relevant by comparing your methods to those used by the NAPA study on climate social vulnerability (Min of Environment 2010)

Please also note the supplement to this comment:
http://www.nat-hazards-earth-syst-sci-discuss.net/nhess-2017-137/nhess-2017-137-RC1-supplement.pdf

---

## Referee Comment (RC2) · Anonymous Referee #2 · 23 Jul 2017

The paper is interesting and it deals with a relevant topic. However, I think some improvements are required and for this I recommend major revisions. In particular, the introduction of the paper is strongly focused on multi-risk that affected and can, in future, occur again in Nepal. I think a broad susceptibility to the main hazards should be included in the performed analysis, highlighting the district where the multi-hazard is more prone to occur. I think the author could try to map as point data the districts affected by several natural hazards and obtain a final score, which highlights the worst ones. This could be an idea, but I'm sure the author can also find another way to account for this. This evaluation could be coupled with the final map and provide more consistency to the results. In the following same typesetting reviews: Page 1: row 7 remove "are reported in Nepal", row 9 "fulfill", row 17 "seismotectonic" is enough, row

24 remove "in Nepal". Page 2: row 18 remove "in" with "from" Page 3:remove "no any" with "no kind of", row 16 "Table 1" Page 5: Table 2 is not cited

---

## Author Comment (AC1) · 21 Aug 2017

Dear reviewer,

Thanks for your time to review the manuscript and notable comments. I will be revising the language in the revised version. Your concerns would be addressed as below in the revised manuscript:

1. Social vulnerability analysis is kind of package consideration rather than post-eq scenario and multiple disasters are considered while analyzing. It is the dissemination of holistic scenario rather than focusing on the single event. In order to incorporate the post-eq scenario, the seismic risk/vulnerability would be better and I have disseminated other papers related to such aspects too. Apart from this, climate change is the cause

of a disaster and this study is the effect study only. The overall scenario is presented in social vulnerability irrespective of a particular event. You have raised a strong concern of multi-hazards risk assessment, and I am working on that. Meanwhile, a multi-risk analysis is not the scope of this paper.

2. I will opt to provide some criticisms (synthesis) considering your suggestions. 3. The vulnerability formula is well known and the reference is provided. One of the references in the text shows that three different techniques to estimate the vulnerability scores do not lead to variation, thus the generalized one is used in this study. 4. Your comment is noted. But scoring based vulnerability mapping does not exist to the best of my knowledge. I will update and amend the sentence when necessary.

Thank you!

---

## Author Comment (AC2) · 21 Aug 2017

Dear reviewer,

Thank you for your time to review the manuscript. Your suggestions will be thoroughly considered while performing district level multi-hazard scenario depiction. For this, I will prepare additional maps with the available database for major hazards and then compare with the results. I will amend the language for clarity and conciseness too.

Thank you!
* * *

---

## Editor Decision (ED1)

**Assessment of Social Vulnerability to Natural Hazards in Nepal**

Dipendra Gautam[1]

[1]Structural and Earthquake Engineering Research Institute, Kathmandu, Nepal

*Correspondence to*: Dipendra Gautam (dipendra.gautam.seri@gmail.com)

5 **Abstract.** This paper disseminates district-wise social vulnerability to natural hazards in Nepal. Disasters such as earthquake, flood, landslide, epidemic, and drought are common in Nepal. Every year thousands of people are killed and huge economic and environmental losses occur in Nepal due to various natural disasters. Although natural hazards are well recognized, quantitative as well as qualitative social vulnerability mapping does not exist until now in Nepal. This study aims to quantify the social vulnerability in local scale considering all 75 districts using available census. To perform district level vulnerability

10 mapping, 13 variables were selected and aggregated indexes were plotted in Arc GIS environment. The sum of results shows that only 4 districts in Nepal have very low social vulnerability index whereas 46 districts (61%) are under moderate to high social vulnerability level. Vulnerability mapping highlights the immediate need for decentralized frameworks to tackle natural hazards in district level, meanwhile, the results of this study can contribute to preparedness, planning and resource management, inter-district coordination, contingency planning and public awareness efforts.

**1 Introduction**

15 Nepal is characterized by frequent occurrence of natural disasters throughout the territory. Geo-Seismotectonics, annual torrential precipitation, climate change impacts, among others are the leading causes of natural disasters in Nepal. Notably, the first decade of the 21[st] century was contemptible for Nepal due to loss of above 15000 people and other tens of thousands were injured. Apart from this, multi-faceted disasters occur every year leading to enormous losses in socio-economic and

20 environmental sectors. The global vulnerability rank of Nepal as depicted by UNDP/BCPR (2004) depicts that Nepal is 20[th] most multi-hazard prone country; 4[th] in the case of climate change related hazards; 11[th] in the case of earthquake hazard and 30[th] in terms of flood hazards. Recent events such as 2009 flood in eastern Nepal, 2011 earthquake in eastern Nepal, Gorkha earthquake (2015) and 2017 flood justify the occurrence of frequent and devastating events. Although it is well known in Nepal that country is disaster-prone, multi-hazard risk assessment is not performed yet, thus exhaustive and regional scale risk

25 scenario for most parts of the country is not well understood. To this end, risk assessment is crucial for Nepal especially due

to exposure characteristics, frequent disasters and, substandard infrastructural preparedness. The overall risk due to natural hazards that depends on hazard (H), vulnerability (V), and exposure to hazard (E) can be written as follows:

$$R = H{\times}V{\times}E \qquad ... (1)$$

Endorsement of hazards and associated studies started in Nepal since 1982 when the Natural Calamity Relief Act (1982) was formulated for the first time in South Asia and before most of the countries in the world developed their risk reduction strategies. Even though the policy was formulated 1980s, relevant changes in subsequent years were not per expected level thus the same act is functional till date. In the policy level, endorsement of building code act since 2003 became the first major intervention to counteract the earthquake hazard, however, implementation of building code remains confined to few urban centers of Nepal and most parts of the country continue to follow the conventional construction technology till date. The evidence from 2015 Gorkha earthquake also notes that the building collapse was largely confined to rural and suburb neighborhoods of Nepal whereas the urban areas sustained relatively lower damage (for details see: Gautam and Chaulagain 2016; Gautam et al. 2016; Varum et al. 2018). Limited works related to earthquake and landslide hazard mapping have been done from local to regional scales in Nepal. Chaulagain et al. (2015) assessed seismic risk and mapped the seismic hazard across Nepal. Similarly, Paudyal et al. (2012), Gautam and Chamlagain (2016) and Gautam et al. (2017) performed local scale hazard analyses and developed microzonation maps. In addition to this, Chaulagain et al. (2016) performed loss estimation assessment in case of earthquakes for Kathmandu valley. In of the case of landslides, Devkota et al. (2013) developed landslide susceptibility maps in regional scale whereas studies related to other catchments and regions are not common in literature. After 2000, earthquake is widely discussed topic in Nepal in policy to the local level. Meanwhile, landslides, floods, and other hazards are not given equal emphasis in policy level and academic researches. The earthquake risk in Nepal is distributed throughout Nepal so every community has the potential of similar exposure if structural vulnerabilities are not considered. Meanwhile, the epidemic risk is also distributed throughout Nepal. Two distinct and more localized hazards in Nepal are the landslides and the floods. Based on the previous events, generalized flood and landslide risks are mapped in Figs. 1 and 2 respectively.

None to the best of author's knowledge has covered social vulnerabilit to natural hazards even though risk perception has reached up to the public level and awareness is exponentially increased in almost all settlements of Nepal. It is worthy to note here, the awareness noted above is limited to earthquake hazard only, whereas, other hazards are not perceived as devastating as the earthquake in public level. Centralized and urban-concentrated resource allocation practice is still becoming perilous to the public of remote locations of Nepal as reinforced by the evidence after the Gorkha earthquake. During the Gorkha earthquake, people in the remote locations were not reached for several weeks after the main shock and whereabouts of thousands of people was unknown for many days. Most of the urban as well as rural settlements are exposed to multi-hazards, in this context, social vulnerability analysis, and mapping is immediately needed for Nepal. Such mapping can have direct influence in policy-making to preparedness activities. Apart from this, even ordinary people could perceive the level of vulnerability from the map so awareness activities could be effectively launched.

[Figure]

**Fig. 1** Generalized flood risk map for Nepal

Social vulnerability analysis in terms of estimated indexes considering a number of variables is widely practiced since the late 1980s. For example, Blaikie and Brookfield (1987), Chambers (1989), Dahl (1991), Cutter et al. (1997), Balaikie et al. (1994), Mileti (1999), Morrow (1999), King and MacGregor (2000) and Cutter et al. (2003) among others provided strong background and motivation for development and implication of social vulnerability index. After 2005, intensive focus has been provided in construction and mapping of social vulnerability index (e.g. de Oliviera Mendes (2009), Wood et al. (2010), Bjarnadottir et al. (2011), Holand et al. (2011), Yoon (2012), Armas and Gavris (2013), Lixin et al. (2013), Guillard-Gonçalves et al. (2013), Siagian et al. (2014), Garbutt et al. (2015), Hou et al. (2015), de Loyola Hummell et al. (2016), Frigerio and de Amicis (2016), Roncancio and Nardocci (2016)). On the contrary, limited work is done in Nepal, however, that is limited to climate change vulnerability only (GoN 2010), even though natural hazards are frequent due to the tectonic setting, annual torrential precipitation, steep topography, climate change and unsustainable and haphazard construction practices as well as due to lack of basic health facilities. In addition, Nepal's preparedness and policy interventions are far below when the existing hazard, exposure, and perception level is considered; that is leading to enormous losses every year. To fulfill the gap between exposure and preparedness, this study depicts district level social vulnerability mapping based on vulnerability scores calculated from selected variables.  After all, some suggestions are made for policy, preparedness, and future interventions.

[Figure]

**Fig. 2** Generalized landslide risk map for Nepal

**2 Materials and Methods**

Nepal does not update the database for population, households, infrastructures, facilities, and others every year. Moreover,

5    digital database is limited thus only the census is the reliable data source to obtain the database for several socioeconomic variables. Even in the case of the census, the coverage in terms of variables is largely constrained to population categories thus more specific data like single year population, per capita income in the local level, village level census is still lacking until 2011 census however although 2011 census progressed appreciably in comparison to the 2001 census. The present study is based on 2011 census as reported by the Central Bureau of Statistics (CBS), National Planning Commission (CBS 2012). Both

[revised manuscript text omitted]

10   isolation of most of the mountainous districts specifically in the western mountain region. Similarly, the cellular phone service was opened to the public only after 2006 and this facility was concentrated to major urban centers and southern Indo-Gangetic plains until 2010. However, in later dates, the reach has become far better as highlighted by the variable (N2). Information and communication is a very important aspect for rapid response and life safety. For example, the 2012 Seti river flood in western Nepal was instantly broadcasted to the downstream people thus the losses were far less than expected. Communication systems

15   in Nepal are also concentrated in urban areas, Indo-Gangetic plains and up to middle mountains leaving behind the high mountains and western mountains far behind thus variance is observed to be high. Almost all districts in Nepal have a higher population of females than males. It is associated with the social norms in terms of the importance of son to continue the future generation. This concept has been eradicated in major urban centers however southern plains and mountains have not progressed much hence female population is very high in such areas.  The high female population is partly also due to the

20   status of education in eastern and central mountains where people have better access to family planning tools thus they started to give birth to a fewer child than the western mountain peoples. The sparse distribution of the population in mountains is surprisingly low due to migration towards the areas with better facilities. In case of eastern and central mountains, the population change between 2000 to 2010 is negative leading to negative population growth rate. The armed conflict was at its peak during 2000 to 2006 thus people migrated to urban areas where security was assured. Due to the social provisions imposed

25   by the rebels, the western mountain districts did not follow the same trend as that of eastern mountains thus western mountains maintained positive population growth. Kathmandu district (serial 31 in Fig. 3) has the population density of 4416 people per square kilometers. On the contrary, Manang district (serial 37 in Fig. 3) has population density of 3 people per square kilometers. Percentage of female-headed population in Nepal started increasing after 2000 due to change in the social norms that were associated with male's supremacy. The Government of Nepal offered discounts on land title transfer tax if the title

30   is changed to females thus the trend is increasing since then. Natural disasters have historically suffered females the most in

Nepal (Chaulagain et al. 2018). For example, Gorkha earthquake of 2015 April 25 killed more females than males. This is because females in Nepal are mostly confined to household activities and remain inside their houses during the disasters. Similar observations were made during the floods in the southern plains at various times.

**Table 3.** Descriptive statistics of variables considered for social vulnerability assessment

| Variables | Standard deviation | Mean | Max. | Min. |
|-----------|--------------------|------|------|------|
| N1 | 5.45 | 95.18 | 99.39 | 69.62 |
| N2 | 4.24 | 12.17 | 22.66 | 3.93 |
| N3 | 10.96 | 18.53 | 45.79 | 1.66 |
| N4 | 2.34 | 51.98 | 56.81 | 44 |
| N5 | 585.69 | 312 | 4416 | 3 |
| N6 | 4.28 | 8.67 | 17.7 | 1.20 |
| N7 | 0.62 | 4.90 | 6.44 | 3.92 |
| N8 | 8.96 | 32.14 | 54.35 | 12.14 |
| N9 | 14.80 | 9.72 | 61.23 | -31.8 |
| N10 | 0.90 | 2.43 | 5.39 | 0.9 |
| N11 | 4.69 | 44.48 | 52.79 | 29.8 |
| N12 | 20.84 | 39.77 | 79.26 | 0.85 |
| N13 | 25.73 | 42.98 | 95.98 | 1.88 |

The average number of people per household is varied in Nepal mainly due to geographical locations and cultural groups. In remote locations, the child birth rate is usually high thus the average number of people per household is high. However, in the case of urban neighborhoods, multiple families share a single building (either joint family or rented family). The census lacks specific information regarding the rented families thus it was not possible to classify and define a separate variable for this
10 aspect.

Nepal has progressed considerably in education sector especially after 2000. Students from marginalized communities, ethnic minorities, certain geographical locations are given stipends and reservations for the basic and higher education. Although the educational status is not comparable to developed states till now, people who could read and write in the Nepali language are defined as literate in Nepal. Per the 2011 census, literacy status in some of the southern plain districts and western mountain
15 districts was very low. This is due to social problems like the value of education in the community, early marriage and high dependence on subsistence farming. Population with at least one deficiency varies between 0.9 to 5.39% in Nepal. Nepal is striving for basic health facilities until now. Majority of the health facilities, preventive measures, and child vaccinations were started after the restoration of democracy in 1990 thus still a considerable fraction of the population has health deficiencies.

Meanwhile. til now, Nepal eradicated Polio and Malaria and progressed in controlling other diseases too. The variation of the economically unproductive population (population below 15 years and above 60 years) ranges from 29.8 to 52.79% in Nepal. This denotes dependent population is very high and impact of a disaster is particularly intense in such groups. Sanitation is still a big issue for Nepalese people. Percentage of households without toilets are still up to 79.26%. In addition to this, clean

5 drinking water is not assured to each household in Nepal. The various sources of water supply like tap water, springs and others are not independently verified in terms of water quality. Apart from this, thousands of people suffering from water-borne diseases are reported during every spring and monsoon in Nepal. The 2011 census consists various water supply resources for households in Nepal, however, due to water quality issue, this variable was not considered in this study. The social vulnerability indexes for each district per generalized conventions based on standard deviation are depicted in Fig. 4.

10 As shown in Fig. 4, social vulnerability to natural hazards is higher in western mountains than the eastern and central regions of Nepal. Similarly, only four districts are under very low vulnerability level in Nepal. This scenario depicts higher vulnerability to natural hazards nationwide. Western Nepal is long identified as the potential hotspot of future mega-earthquake, famine and epidemics thus instant interventions are required to tackle the very high to high vulnerability status of districts in this region.

[Figure]

**Fig. 4** District-wise social vulnerability to natural hazards in Nepal

The frequency of districts in terms of social vulnerability level is outlined in Table 4. As shown in Table 4, 6 (8%) out of 75 districts are under very high social vulnerability level, 18 (24%) districts are under high vulnerability level, 22 (29.33%) districts are under moderate social vulnerability level, 25(33.33%) districts are under low vulnerability level, and 4 (5.34%) districts are under very low vulnerability level in Nepal. Results show that 32% of districts are under high vulnerability level thus District Natural Disaster Relief Committee (DDNRC) needs to be strengthened with adequate resources in these districts more than others. As observed during the Gorkha earthquake of 2015, Sindhupalchowk landslide of 2014 and Koshi flood of 2008; every relief, response, and recovery effort was governed by the Central Natural Disaster Relief Committee (CNDRC) that led in delayed response and the efforts were sometimes hindered by the weather extremities too. In addition to this, it was observed that Local Natural Disaster Relief Committee (LNDRC) were completely defunct during these events thus preparedness in terms of uplifting local committees is immediately needed in Nepal. Even after the federal states become functional, districts will not be changed thus contingency planning to sustainable natural disaster preparedness initiatives are urgent especially for western mountains. Resource allocations, training first responders, district level planning and overall budget allocation can have the benefit of the mapping done in this study. In addition to this, one door policy and coordinated response mechanisms as highlighted by Gautam (2018) could be formulated in the highly vulnerable districts. As shown in Fig. 1 and Fig. 2, and considering the throughout distribution of the earthquake and epidemic risks, the higher social vulnerability level of southern plains of Nepal could be attributed to flood risk partly. Whereas, the vulnerability in the case of middle and high mountains could have been associated with the landslide risk. However, to present the exact impacts of each hazard, social vulnerability due to individual hazard should be considered.

**Table 4.** Frequency of districts in terms of social vulnerability level

| Level of social vulnerability | Number of districts | Districts |
| --- | --- | --- |
| Very high | 6 | Jajarkot, Kalikot, Mugu, Humla, Bajura, Bajhang |
| High | 18 | Siraha, Saptari, Mahottari, Sarlahi, Sindhuli, Rautahat, Bara, Kapilvastu, Rolpa, Rukum, Salyan, Dailekh, Dolpa, Jumla, Achham, Doti, Baitadi, Darchula |
| Moderate | 22 | Taplejung, Morang, Bhojpur, Solukhumbu, Okhaldhunga, Khotang, Udaypur, Dhanusha, Ramechhap, Rasuwa, Makwanpur, Parsa, Myagdi, Rupandehy. Pyuthan, Dang, Banke, Bardiya, Surkhet, Kailali, Kanchanpur, Dadeldhura |
| Low | 25 | Panchthar, Ilam, Jhapa, Sunsari, Dhankuta, Tehrathum, Sankhuwasabha, Dolakha, Sindhupalchowk, Kavre, Kathmandu, Nuwakot, Dhading, Chitwan, Gorkha, Lamjung, Tanahun, Kaski, Mustang, Parbat, Baglung, Gulmi, Palpa, Nawalparasi, Arghakhachi |

Being a multi-hazard prone country, multi-hazard risk assessment is urgently needed so that social vulnerability mapping could be integrated with the multi-hazard maps to depict precise thematic maps. Nepal lacks highly classified researches regarding hazard mapping, however, the current focus is not sufficient to develop reliable outputs thus more integrated efforts from the government, as well as researchers, is needed. It is obvious that if national priority is considered in specific hazard to multi-hazard mapping across the country, responding natural disasters would be much easier in terms of policies to ad-hoc interventions. In addition to this, data management, digitization, and coverage of more variables during census will increase the quality of social vulnerability indexes, thus future censuses should consider more variables. Finally, local constituencies below the district level are being formulated as the primary units of the Federal Democratic Republic of Nepal and local constituency level social vulnerability mapping will be more effective than district level mapping if the database can be organized immediately after formulation of such constituencies.

**4 Conclusion**

This study is the first attempt to understand the district level vulnerability in Nepal. The social vulnerability score is calculated and mapped for all 75 districts of Nepal. Being a natural disaster-prone country, Nepal needs to develop effective mitigation, prevention and contingency plans for all potential natural hazards so this study could be fundamental for the policy makers and stakeholders to initiate interventions at the district level. The sum of results highlights that western mountain districts are under very high to high social vulnerability status, whereas, eastern and central regions depicted low to moderate social vulnerability to natural disasters in general. Losses due to natural hazards in western mountain districts would be very high in the case of major natural hazards thus immediate actions are needed. Previous natural disasters have reflected a poor coordination, delayed response, and marginal preparedness scenario from the central level. Thus, decentralization in terms of preparedness, response and recovery are necessary for Nepal because of the district wise variation of social vulnerability to natural hazards.

In social vulnerability assessment, data constraint plays the important role thus the results may be varied per the number of variables. Consideration of more variables is important to assure precision and proper representation of social vulnerability. In addition to this, spatial variation within a district has also remarkable influence as vulnerability mapping considers a uniform variation of values of variables within the district that is not strictly per the ground condition. To overcome this, the local level social vulnerability mapping should be considered in the future. Apart from this, exhaustive and more reliable social vulnerability index (SoVI) mapping and integrated multi-hazard risk assessment based on principal component analysis is needed for Nepal.

---

## Author Response (AR2)

11/10/17

Dear Professor Fuchs,

I would like to thank you and the reviewers for the time and efforts to review the manuscript.

Per your suggestion to incorporate the second reviewer's concerns, I have modified the manuscript accordingly.

The issues raised by the reviewer are highlighted by green colored text in tracked manuscript. Following changes/explanations are made per the reviewer's comments:

PAGE-1

1. Line 5-6: earthquakeS, floodS, landslideS, epidemicS
   Re: Changes are made in lineas 5-6.
2. Line 7: it is currently considered that natural disasters do not exist, they are a consequence of management, Ok to natural hazards
   Re: Change is made in line 7.
3. Line 17: wrong use of word
   Re: The sentence is restructured.
4. Line 20: twice depicts in sentence
   Re: One depicts is replaced by "highlights"
5. Line 23: explain/ illustrate/ demonstrate, not justify
   Re: Change is made.

PAGE-2

1. Line 24: please social vulnerability as compared to other types of vulnerability, add references with regards to how it appeared in literature. it is important to note whether you consider social vulnerability to be independent or dependent on the type of hazard...
   Re: Comparison with other vulnerabilities in not the scope of the work. Meanwhile, social vulnerability to natural hazards is considered in the study independent on the type of hazard. This study relies on the same approach that the other scientists are practicing in the other part of the world.

PAGE-3
1. Line 1: source?
   Re: I mapped the past events per the suggestion of one of the reviewer in the earlier version of the manuscript. So, no source is needed as it is not published by anyone else.

PAGE-4
1. Line 1: the map doesnt tell us much, except that most of Nepal is affected...
ICIMOD has put out several recent maps that would be more effective... and source is missing
   Re: I mapped the previous events myself and depicted the district level risk map so no source is needed. I checked ICIMOD maps, but those were localized ones thus I followed the suggestions of one of the reviewer in the previous version of the manuscript.

PAGE-7:
The section is revised per the reviewer's suggestion.

Line 29: this observation on female casualties and disasters belongs to the general introduction, not in the section which analyses the district-level data
Re: The sentence is removed.
Line 30: Revise sentence structure
Re: The sentence is revised and now is in lines 25-26, page 8.

PAGE-8:
Line 6: remove is, replace with varies
Re: Change is made (page 8, line 8).

PAGE-9:
Line 1: review
Re: Change is made (page 7 line 14).

Map: very large difference between Mustang and neighboring district Dolpa... Why is it so, please add explanation
Re: The mapping is based on 2011 census, it is well explained in the discussion section.

PAGE-10:
Line 3: be consistent, are two decimal points really needed?
Re: Only one digit after decimal is used. Amendments are made.

Line 9: was
Re: Change is made.

I would like to thank you Professor Sven Fuchs and the reviewers for constructive feedbacks.

Best regards,
Dipendra

**Assessment of Social Vulnerability to Natural Hazards in Nepal**

Dipendra Gautam[1]

[1]Structural and Earthquake Engineering Research Institute, Kathmandu, Nepal

*Correspondence to*: Dipendra Gautam (dipendra.gautam.seri@gmail.com)

5 **Abstract.** This paper disseminates district-wise social vulnerability to natural hazards in Nepal. Disasters such as earthquakes, floods, landslides, epidemics, and droughts are common in Nepal. Every year thousands of people are killed and huge economic and environmental losses occur in Nepal due to various natural hazards. Although natural hazards are well recognized, quantitative as well as qualitative social vulnerability mapping does not exist until now in Nepal. This study aims to quantify the social vulnerability in local scale considering all 75 districts using available census. To perform district level vulnerability

10 mapping, 13 variables were selected and aggregated indexes were plotted in Arc GIS environment. The sum of results shows that only 4 districts in Nepal have very low social vulnerability index whereas 46 districts (61%) are under moderate to high social vulnerability level. Vulnerability mapping highlights the immediate need for decentralized frameworks to tackle natural hazards in district level, meanwhile, the results of this study can contribute to preparedness, planning and resource management, inter-district coordination, contingency planning and public awareness efforts.

**1 Introduction**

Nepal is characterized by frequent occurrence of natural disasters throughout the territory. Geo-Seismotectonics, annual torrential precipitation, climate change impacts, among others are the leading causes of natural disasters in Nepal. Notably, the first decade of the 21st century, Nepal observed loss of above 15000 people and other tens of thousands of injuries. Apart from this, multi-faceted disasters occur every year leading to enormous losses in socio-economic and environmental sectors. The

20 global vulnerability rank of Nepal as depicted by UNDP/BCPR (2004) highlights that Nepal is 20th most multi-hazard prone country; 4th in the case of climate change related hazards; 11th in the case of earthquake hazard and 30th in terms of flood hazards. Recent events such as 2009 flood in eastern Nepal, 2011 earthquake in eastern Nepal, Gorkha earthquake (2015) and 2017 flood illustrate the occurrence of frequent and devastating events. Although it is well known in Nepal that country is disaster-prone, multi-hazard risk assessment is not performed yet, thus exhaustive and regional scale risk scenario for most

25 parts of the country is not well understood. To this end, risk assessment is crucial for Nepal especially due to exposure

characteristics, frequent disasters and, substandard infrastructural preparedness. The overall risk due to natural hazards that depends on hazard (H), vulnerability (V), and exposure to hazard (E) can be written as follows:

$$R = H \times V \times E \qquad ...(1)$$

Endorsement of hazards and associated studies started in Nepal since 1982 when the Natural Calamity Relief Act (1982) was formulated for the first time in South Asia and before most of the countries in the world developed their risk reduction strategies. Even though the policy was formulated 1980s, relevant changes in subsequent years were not per expected level thus the same act is functional till date. In the policy level, endorsement of building code act since 2003 became the first major intervention to counteract the earthquake hazard, however, implementation of building code remains confined to few urban centers of Nepal and most parts of the country continue to follow the conventional construction technology till date. The evidence from 2015 Gorkha earthquake also notes that the building collapse was largely confined to rural and suburb neighborhoods of Nepal whereas the urban areas sustained relatively lower damage (for details see: Gautam and Chaulagain 2016; Gautam et al. 2016; Varum et al. 2018). Limited works related to earthquake and landslide hazard mapping have been done from local to regional scales in Nepal. Chaulagain et al. (2015) assessed seismic risk and mapped the seismic hazard across Nepal. Similarly, Paudyal et al. (2012), Gautam and Chamlagain (2016) and Gautam et al. (2017) performed local scale hazard analyses and developed microzonation maps. In addition to this, Chaulagain et al. (2016) performed loss estimation assessment in case of earthquakes for Kathmandu valley. In of the case of landslides, Devkota et al. (2013) developed landslide susceptibility maps in regional scale whereas studies related to other catchments and regions are not common in literature. After 2000, earthquake is widely discussed topic in Nepal in policy to the local level. Meanwhile, landslides, floods, and other hazards are not given equal emphasis in policy level and academic researches. The earthquake risk in Nepal is distributed throughout Nepal so every community has the potential of similar exposure if structural vulnerabilities are not considered. Meanwhile, the epidemic risk is also distributed throughout Nepal. Two distinct and more localized hazards in Nepal are the landslides and the floods. Based on the previous events, generalized flood and landslide risks are mapped in Figs. 1 and 2 respectively.

None to the best of author's knowledge has covered social vulnerability to natural hazards even though risk perception has reached up to the public level and awareness is exponentially increased in almost all settlements of Nepal. It is worthy to note here, the awareness noted above is limited to earthquake hazard only, whereas, other hazards are not perceived as devastating as the earthquake in public level. Centralized and urban-concentrated resource allocation practice is still becoming perilous to the public of remote locations of Nepal as reinforced by the evidence after the Gorkha earthquake. During the Gorkha earthquake, people in the remote locations were not reached for several weeks after the main shock and whereabouts of thousands of people was unknown for many days. Most of the urban as well as rural settlements are exposed to multi-hazards, in this context, social vulnerability analysis, and mapping is immediately needed for Nepal. Such mapping can have direct influence in policy-making to preparedness activities. Apart from this, even ordinary people could perceive the level of vulnerability from the map so awareness activities could be effectively launched.

[Figure]

**Fig. 1** Generalized flood risk map for Nepal

Social vulnerability analysis in terms of estimated indexes considering a number of variables is widely practiced since the late 1980s. For example, Blaikie and Brookfield (1987), Chambers (1989), Dahl (1991), Cutter et al. (1997), Balaikie et al. (1994),
5   Mileti (1999), Morrow (1999), King and MacGregor (2000) and Cutter et al. (2003) among others provided strong background and motivation for development and implication of social vulnerability index. After 2005, intensive focus has been provided in construction and mapping of social vulnerability index (e.g. de Oliviera Mendes (2009), Wood et al. (2010), Bjarnadottir et al. (2011), Holand et al. (2011), Yoon (2012), Armas and Gavris (2013), Lixin et al. (2013), Guillard-Gonçalves et al. (2013), Siagian et al. (2014), Garbutt et al. (2015), Hou et al. (2015), de Loyola Hummell et al. (2016), Frigerio and de Amicis (2016),
10   Roncancio and Nardocci (2016)). On the contrary, limited work is done in Nepal, however, that is limited to climate change vulnerability only (GoN 2010), even though natural hazards are frequent due to the tectonic setting, annual torrential precipitation, steep topography, climate change and unsustainable and haphazard construction practices as well as due to lack of basic health facilities. In addition, Nepal's preparedness and policy interventions are far below when the existing hazard, exposure, and perception level is considered; that is leading to enormous losses every year. To fulfill the gap between exposure
15   and preparedness, this study depicts district level social vulnerability mapping based on vulnerability scores calculated from selected variables. After all, some suggestions are made for policy, preparedness, and future interventions.

[Figure]

**Fig. 2** Generalized landslide risk map for Nepal

**2 Materials and Methods**

Nepal does not update the database for population, households, infrastructures, facilities, and others every year. Moreover,
5   digital database is limited thus only the census is the reliable data source to obtain the database for several socioeconomic
variables. Even in the case of the census, the coverage in terms of variables is largely constrained to population categories thus
more specific data like single year population, per capita income in the local level, village level census is still lacking until
2011 census however although 2011 census progressed appreciably in comparison to the 2001 census. The present study is
based on 2011 census as reported by the Central Bureau of Statistics (CBS), National Planning Commission (CBS 2012). Both
10  2011 and 2001 census were used to estimate the 13 variables used in this study. Only 13 variables were used in this study
considering the reliable and available ones as most of the information were not strictly associated with social vulnerability to
natural hazards. Table 1 depicts the description of variables used in this study along with cardinality. Broadly, social
vulnerability assessment can be categorized under two approaches as: a) deductive and b) inductive. Deductive approach is
based on the selection of limited variables as done by Cutter et al. (2000), Wu et al. (2002), Zahran et al. (2008) and others.
15  Meanwhile, inductive approach uses more organized and exhaustive social vulnerability assessment framework with all
possible variants considered at a time. Recent advances in social vulnerability assessment are more focused towards inductive

[revised manuscript text omitted]
 high female population is partly also due to the status of education in eastern and central mountains where people have better access to family planning tools thus they started to give birth to a fewer child than the western mountain peoples. The sparse distribution of the population in mountains is surprisingly low due to migration towards the areas with better facilities. In case of eastern and central mountains, the population change between 2000 to 2010 is negative leading to negative population growth rate. The armed conflict was at its peak during 2000 to 2006 thus people migrated to

20    urban areas where security was assured. Due to the social provisions imposed by the rebels, the western mountain districts did not follow the same trend as that of eastern mountains thus western mountains maintained positive population growth. Kathmandu district (serial 31 in Fig. 3) has the population density of 4416 people per square kilometers. On the contrary, Manang district (serial 37 in Fig. 3) has population density of 3 people per square kilometers. Percentage of female-headed population in Nepal started increasing after 2000 due to change in the social norms that were associated with male's supremacy.

25    Per the recent study conducted by Chaulagain et al. (2018), women are affected more by the earthquakes than the men in the case of every notable earthquakes in Nepal. For example, Gorkha earthquake of 2015 April 25 killed more females than males. This is because females in Nepal are mostly confined to household activities and remain inside their houses during the disasters. Similar observations were made during the floods in the southern plains at various times.

**Table 3.** Descriptive statistics of variables considered for social vulnerability assessment

| Variables | Standard deviation | Mean | Max. | Min. |
|---|---|---|---|---|
| N1 | 5.45 | 95.18 | 99.39 | 69.62 |
| N2 | 4.24 | 12.17 | 22.66 | 3.93 |

| | | | |
|---|---|---|---|
| N3 | 10.96 | 18.53 | 45.79 | 1.66 |
| N4 | 2.34 | 51.98 | 56.81 | 44 |
| N5 | 585.69 | 312 | 4416 | 3 |
| N6 | 4.28 | 8.67 | 17.7 | 1.20 |
| N7 | 0.62 | 4.90 | 6.44 | 3.92 |
| N8 | 8.96 | 32.14 | 54.35 | 12.14 |
| N9 | 14.80 | 9.72 | 61.23 | -31.8 |
| N10 | 0.90 | 2.43 | 5.39 | 0.9 |
| N11 | 4.69 | 44.48 | 52.79 | 29.8 |
| N12 | 20.84 | 39.77 | 79.26 | 0.85 |
| N13 | 25.73 | 42.98 | 95.98 | 1.88 |

[Figure]

**Fig. 4** District-wise social vulnerability to natural hazards in Nepal

The frequency of districts in terms of social vulnerability level is outlined in Table 4. As shown in Table 4, 6 (8%) out of 75 districts are under very high social vulnerability level, 18 (24%) districts are under high vulnerability level, 22 (29.3%) districts

are under moderate social vulnerability level, 25(33.3%) districts are under low vulnerability level, and 4 (5.3%) districts are under very low vulnerability level in Nepal. Results show that 32% of districts are under high vulnerability level thus District Natural Disaster Relief Committee (DDNRC) needs to be strengthened with adequate resources in these districts more than others. As observed during the Gorkha earthquake of 2015, Sindhupalchowk landslide of 2014 and Koshi flood of 2008; every

5    relief, response, and recovery effort was governed by the Central Natural Disaster Relief Committee (CNDRC) that led in delayed response and the efforts were sometimes hindered by the weather extremities too. In addition to this, it was observed that Local Natural Disaster Relief Committee (LNDRC) was completely defunct during these events thus preparedness in terms of uplifting local committees is immediately needed in Nepal. Even after the federal states become functional, districts will not be changed thus contingency planning to sustainable natural disaster preparedness initiatives are urgent especially for

10   western mountains. Resource allocations, training first responders, district level planning and overall budget allocation can have the benefit of the mapping done in this study. In addition to this, one door policy and coordinated response mechanisms as highlighted by Gautam (2018) could be formulated in the highly vulnerable districts. As shown in Fig. 1 and Fig. 2, and considering the throughout distribution of the earthquake and epidemic risks, the higher social vulnerability level of southern plains of Nepal could be attributed to flood risk partly. Whereas, the vulnerability in the case of middle and high mountains

15   could have been associated with the landslide risk. However, to present the exact impacts of each hazard, social vulnerability due to individual hazard should be considered.

Table 4. Frequency of districts in terms of social vulnerability level

| Level of social vulnerability | Number of districts | Districts |
|---|---|---|
| Very high | 6 | Jajarkot, Kalikot, Mugu, Humla, Bajura, Bajhang |
| High | 18 | Siraha, Saptari, Mahottari, Sarlahi, Sindhuli, Rautahat, Bara, Kapilvastu, Rolpa, Rukum, Salyan, Dailekh, Dolpa, Jumla, Achham, Doti, Baitadi, Darchula |
| Moderate | 22 | Taplejung, Morang, Bhojpur, Solukhumbu, Okhaldhunga, Khotang, Udaypur, Dhanusha, Ramechhap, Rasuwa, Makwanpur, Parsa, Myagdi, Rupandehy. Pyuthan, Dang, Banke, Bardiya, Surkhet, Kailali, Kanchanpur, Dadeldhura |
| Low | 25 | Panchthar, Ilam, Jhapa, Sunsari, Dhankuta, Tehrathum, Sankhuwasabha, Dolakha, Sindhupalchowk, Kavre, Kathmandu, Nuwakot, Dhading, Chitwan, Gorkha, Lamjung, Tanahun, Kaski, Mustang, Parbat, Baglung, Gulmi, Palpa, Nawalparasi, Arghakhachi |
| Very low | 4 | Lalitpur, Bhaktapur, Manang, Syangja |

Being a multi-hazard prone country, multi-hazard risk assessment is urgently needed so that social vulnerability mapping could be integrated with the multi-hazard maps to depict precise thematic maps. Nepal lacks highly classified researches regarding hazard mapping, however, the current focus is not sufficient to develop reliable outputs thus more integrated efforts from the government, as well as researchers, is needed. It is obvious that if national priority is considered in specific hazard to multi-hazard mapping across the country, responding natural disasters would be much easier in terms of policies to ad-hoc interventions. In addition to this, data management, digitization, and coverage of more variables during census will increase the quality of social vulnerability indexes, thus future censuses should consider more variables. Finally, local constituencies below the district level are being formulated as the primary units of the Federal Democratic Republic of Nepal and local constituency level social vulnerability mapping will be more effective than district level mapping if the database can be organized immediately after formulation of such constituencies.

**4 Conclusion**

This study is the first attempt to understand the district level vulnerability in Nepal. The social vulnerability score is calculated and mapped for all 75 districts of Nepal. Being a natural disaster-prone country, Nepal needs to develop effective mitigation, prevention and contingency plans for all potential natural hazards so this study could be fundamental for the policy makers and stakeholders to initiate interventions at the district level. The sum of results highlights that western mountain districts are under very high to high social vulnerability status, whereas, eastern and central regions depicted low to moderate social vulnerability to natural disasters in general.  Losses due to natural hazards in western mountain districts would be very high in the case of major natural hazards thus immediate actions are needed. Previous natural disasters have reflected a poor coordination, delayed response, and marginal preparedness scenario from the central level. Thus, decentralization in terms of preparedness, response and recovery are necessary for Nepal because of the district wise variation of social vulnerability to natural hazards.

In social vulnerability assessment, data constraint plays the important role thus the results may be varied per the number of variables. Consideration of more variables is important to assure precision and proper representation of social vulnerability. In addition to this, spatial variation within a district has also remarkable influence as vulnerability mapping considers a uniform variation of values of variables within the district that is not strictly per the ground condition. To overcome this, the local level social vulnerability mapping should be considered in the future. Apart from this, exhaustive and more reliable social vulnerability index (SoVI) mapping and integrated multi-hazard risk assessment based on principal component analysis is needed for Nepal.